# The Effect of Deep Micro Vibrotactile Stimulation on Cognitive Function of Mild Cognitive Impairment and Mild Dementia

**DOI:** 10.3390/ijerph19073803

**Published:** 2022-03-23

**Authors:** Ayuto Kodama, Yasuhiro Suzuki, Kazuki Sakuraba, Yu Kume, Hidetaka Ota

**Affiliations:** 1Advanced Research Center for Geriatric and Gerontology, Akita University, Akita 010-8543, Japan; ay-kodama@med.akita-u.ac.jp; 2Department of Complex Systems Science, Graduate School of Information Science, Nagoya University, Nagoya 464-8601, Japan; ysuzuki@nagoya-u.jp; 3Home-Visit Nursing Station Gotenmari, Yurihonjo 018-1301, Japan; syakenbo@gmail.com; 4Department of Occupational Therapy, Graduate school of Medicine, Akita University, Akita 010-8543, Japan; kume.yuu@hs.akita-u.ac.jp

**Keywords:** deep micro vibrotactile, elderly, cognitive function, dementia

## Abstract

Background: The purpose of this study was to clarify the effect of Deep Micro Vibrotactile (DMV) stimulation on the cognitive functions in elderly people with mild cognitive impairment or mild dementia. Methods: A total of 35 participants with dementia from three nursing homes, who had completed treatment with DMV stimulation at 15–40 Hz (hereinafter, 15–40 Hz DMV stimulation) for a month were recruited for this study. The subjects had received continuous 15–40 Hz DMV stimulation for 24 h a day for 1 month. We assessed the effect of the treatment on the cognitive functions (by the word list memory (WM) test, trail making test-part A (TMT-A) and part B (TMT-B), and symbol digit substitution task (SDST)) and physical functions (grip strength (GS) and usual walking speed (UWS)), by comparing the results at the baseline and after the 1-month intervention (DMV stimulation). Results: The results revealed that the performances in the WM test (*p* < 0.05), TMT-B (*p* < 0.05), and SDST (*p* < 0.01) improved significantly after the intervention. Conclusion: Our findings suggest that 15–40 Hz DMV stimulation is might be effective for improving the cognitive functions in elderly people with dementia. Furthermore, our novel findings showed the different effectiveness of the treatment depending on the stage of cognitive impairments.

## 1. Introduction

With the ageing of the population in Japan, the prevalence of dementia is increasing at an alarming rate. People with dementia experience decline in social and occupational functioning, and with disease progression, these patients become unable to take care of themselves more than 6 months, requiring (a) ongoing support for even the simplest activities of daily living, and (b) specific forms of support to replace abnormal behaviors (such as wandering) with more socially acceptable ones [1,2]. The initial symptoms tend to worsen over time. They may be restricted to new and unfamiliar environments at first, eventually spreading to even familiar environments, with impaired spatial and geographical orientation and negative effects on the autonomy, independence, and self-confidence [3,4]. Alzheimer′s disease (AD) is associated with a decline in the cognitive function of the brain and cognitive dysfunction, and is the most common cause of dementia. The histopathological features of AD include extracellular aggregates of amyloid-β (Aβ) peptide and tau-related pathology [5], neural circuits involved in higher cognitive functions eventually becoming disrupted. However, it is considered possible to reduce the pathology of AD by modulating the neural activity [6,7,8,9]. Despite the recent failures of pharmacological treatments targeting Aβ, the amyloid hypothesis is still one of the leading theories put forth to explain the pathogenesis of AD [10]. On the other hand, several studies have examined non-pharmacological treatments (e.g., cognitive training, music therapy, aromatherapy, animal therapy, acupuncture, and chiropractic) for AD, and these avenues are expected to continue to be explored in future research [11,12]. Vieira et al. investigated the effects of light and sound stimulation in 37 AD patients [13]. Visual stimulation was provided by a strobe LED lamp attached to the inner lens of sunglasses, and auditory stimulation was provided through binaural beats in headphones. The two sources of stimulation were combined to provide stimulation at specific frequencies in the range of 1 to 30 Hz. The results showed an improvement in delayed memory in the subject group as compared with the control group. It was speculated that this therapeutic effect may be the result of a “neuroplasticity process”, in which the brain regenerates in response to stimulation. Furthermore, other research has shown that coherent 40 Hz neural oscillations are the fundamental frequency for healthy brain activity and communication in the brain [14,15]. In the study by Clements-Cortes A et al. (2016), both 40-Hz Deep Micro Vibrotactile (DMV) stimulation and visual stimulation with Digital Versatile Discs were applied as interventions for mild AD patients and outpatients in a healthcare facility [16]. This study reported that 40-Hz DMV stimulation improved male functioning in patients with mild AD, although it was still unclear if 15–40 Hz DMV stimulation might have a positive effect on any of the cognitive domains in elderly people with dementia. In our previous study, we reported that DMV stimulation at 15–40 Hz may be useful as one of the non-pharmacological treatments to enhance short-term memory in AD patients, however, it was quite a small-sample size (five participants) [17]. In this study, we recruited a higher number of participants to investigate the effects of DMV stimulation. Thus, the purpose of this study was to clarify the effects of 15–40 Hz DMV stimulation on the cognitive functions in elderly people with MCI or mild dementia with given conditions.

## 2. Materials and Methods

### 2.1. Participants

Subjects for this study were recruited from three nursing homes, after obtaining their informed consent for participation, between September 2020 and July 2021. The target population was elderly people aged 65 years or over who were using Japanese welfare services for elderly people. Subjects with mood/anxiety disorders, mental retardation, and/or severe systemic diseases were excluded. Since the main purpose of this study was to demonstrate a response to DMV stimulation to inform the next proof of concept study, representation of different stages of cognitive dysfunction was crucial. A total sample size of 35 was considered to be adequate to indicate the potential effect resulting from controlled before-and-after trials in this study. Therefore, we conducted a single intervention trial with DMV stimulation for 35 participants.

### 2.2. Procedure for the Assessment

The characteristics of the study subjects were noted from their medical records, including the age, gender, medical history, and medication status. Demographic data of the participants were collected by healthcare workers, including the height, weight, and BMI. The clinical evaluation included determination of the Barthel Index (BI), and assessment by the Neuropsychiatric Inventory-Nursing Home edition (NPI-NH) and Clinical Dementia Rating (CDR) scale. The BI measures the ability of a subject to perform, and the level of dependency in performing the activities of daily living and records (10 items). The total score ranges from 0 (totally dependent) to 100 (totally functionally independent). A high inter-rater reliability of the total score in elderly people has been reported [18]. Evaluation by the NPI-NH was conducted to assess the behavioral and psychological symptoms of dementia (BPSD). The NPI-NH assesses the severity and frequency of 12 different types of behavioral symptoms: delusions, hallucinations, agitation, depression/dysphoria, anxiety, euphoria/elation, apathy/indifference, disinhibition, irritability/lability, aberrant motor behavior, nighttime disturbances, and appetite/eating change. For each symptom, the score can be calculated by multiplying the severity and frequency scores. The total score is the sum of all the symptoms scores [19,20,21]. The CDR scale is a widely used multidimensional measure of intra-individual decline in cognition, behavior and functioning, based on previously achieved competencies in these areas [22]. Furthermore, the CDR has been neuropathologically confirmed, and the content and criterion validity of this scale have been established [23,24]. In the presence of dementia, the CDR can be used to monitor the entire course of the disease, from very mild to severe, avoiding floor and ceiling effects. Based on information obtained from interviews and cognitive functioning tests, each of the six cognitive and functional domains (memory, disorientation, judgement and problem solving, community activities, home and hobbies, and personal care) is rated on a 5-point scale, with 0 representing no impairment, 0.5, 1, 2, and 3 representing very mild, mild, moderate, and severe dementia, respectively. The participants were divided into a mild cognitive impairment (MCI) group (CDR score 0.5) and a mild dementia group (CDR score 1). MCI was defined as follows for this study: (1) the person is neither normal nor has dementia; (2) there is evidence of cognitive deterioration shown by either objectively measured decline over time or subjective reporting of decline by the patient him/herself or informant, in conjunction with objective cognitive deficits; and (3) activities of daily living are preserved and complex instrumental functions are either intact or minimally impaired [25]. This study was conducted with the approval of the ethics committee of the Department of Health Science, Akita University.

### 2.3. Treatment

The DMV system, consisting of an MP3 player (RUIZU^®^ Digital Player X02) and speakers, was used to provide continuous vibrotactile stimulation and inaudible low-frequency DMV stimulation at 15–40 Hz for 24 h a day, for 1 month (YAMAHA-NS-SW050/B). DMV stimulation is a low-frequency stimulation below 40 Hz, that does not affect audible sounds, such as causing sound distortion. Audible sounds, such as speech and environmental sounds, acoustically mask the DMV stimuli and are therefore not heard by the DMV. Strong low-frequency sound pressure can cause vibration of the generator enclosure, windows and doors, whereas DMV is at a sound pressure that does not cause such vibration. Normally, a low-frequency sound of 20 Hz might produce vibrations with a sound pressure of 80 dB or more. Acoustic testing and investing with low-frequency microphones, with or without DMV exposed in the space. In order to carry out acoustic analysis in our study, room sounds with and without DMV were recorded and compared and analyzed using an acoustic analysis software (Figure 1). When mice with AD were exposed to 40 Hz, the accumulation of amyloid-beta was reduced and cognitive functions were improved, suggesting that 40 Hz may inhibit the progression of AD [9]. It has been reported that when rats with AD were exposed to 40 Hz, the accumulation of amyloid-beta in the brain decreased [10], suggesting that 40-Hz stimulation may inhibit the progression of AD. One previous study reported that 40-Hz stimulation was the most effective, one study proposed new use for embedding low frequencies in sound as called the DMV [26]. The vibrotactile stimulation range was set at 15–40 Hz, controlled by an amplifier. These devices were manufactured by Kaga Electronics, Co., LTD (Tokyo, Japan). The stimuli were broadcast from three low-frequency speakers installed at the facility (Figure 2). All participants carried out normal activities of daily living in the DMV-stimulated environment.

### 2.4. Outcome Measures

We assessed the patients for changes in the grip strength (GS) and usual walking speed (UWS) after the intervention. We also evaluated the cognitive status using four cognitive subtests selected from the National Center for Geriatrics and Gerontology Functional Assessment Tool (NCGG-FAT) [27], and the Mini-Mental State Examination (MMSE) [28,29]. The NCGG-FAT is a computerized multidimensional neurocognitive test performed on an iPad with a 9.7-inch touch display (Apple, Cupertino, CA, USA). Assessment by the NCGG-FAT enables evaluation of the effects of interventions on multidimensional cognitive functioning in elderly people. The overall standard is determined from the standard deviation of the measured values with respect to the respective means. The assessment standards are 1, lowest; 2, somewhat low; 3, normal; 4, fine; 5, very fine. The following NCGG-FAT subtests were used (Figure 3).

Subtest 1: WM

The WM test consists of immediate recognition and delayed memory performed using a computer. In the first part, immediate recognition, the subject is asked to memorize 10 words that would appear on a tablet PC. The subjects are then presented with 30 words (10 targets and 20 distractors) and are required to select the 10 target words immediately. This task is repeated three times. The average number of correct answers is scored on a scale of 0 to 10. In the next part of the test, delayed recall, the participants are required to correctly recall the 10 target words after 20 minutes. The responses are assigned a score in the range of 0 to 10. The end result, the sum of the scores in the two tasks, immediate recognition and delayed recall, is calculated.

Subtest 2: TMT-A and Subtest 3: TMT-B

In the TMT-A subtest, participants are instructed to choose the target number as quickly as possible. The target numbers 1 to 15 are displayed in pieces on a screen. The TMT-B consists of selecting the target numbers and characters in sequence.

Subtest 4: SDST

In the SDST, the subjects are shown nine pairs of numbers and symbols at the top of the display. The target symbol is displayed at the center of the display and the selectable numbers are displayed at the bottom. Participants are asked to choose the number corresponding to the target symbol in the center of the panel as quickly as possible. The number of correct answers within 2 minutes is recorded.

### 2.5. Statistical Analysis

In regard to the characteristics of the subjects at the baseline, we used an unpaired t-test to compare the age, gender distribution, height, weight, BMI, BI, and NPI-NH score between the MCI and mild dementia groups. Then, the paired t-test was applied to compare the pre-test/post-test performances of the participants in the GS, UWS, WM, TMT-A and B, SDST, and MMSE.

## 3. Results

According to the CDR, the 35 participants were classified into the MCI group (*n* = 17) and mild dementia group (*n* = 18) (Table 1). The unpaired t-test was used to analyze the differences in the characteristics of the participants between the two groups, and revealed a significant difference in the NPI-NH score between the two groups (*p* < 0.05). However, there were no differences in any of the subject characteristics, such as the age, gender distribution, height, weight, BMI, or BI at the baseline between the two groups.

Next, a comparison of the pre-test and post-test results in the participants is shown in Table 2 and Table 3. The paired t-test was used to analyze the pre-test/post-test differences, and revealed significant differences in the results of the WM test (*p* < 0.05), TMT-B (*p* < 0.05), and SDST (*p* < 0.01). Moreover, we also compared the pre-test/post-test differences in the results between the two groups (Table 3). There were significant differences in the performance in the TMT-B, and SDST in the MCI group, and in the WM test in the mild dementia group at the post-test.

## 4. Discussion

In our previous study, we conducted a preliminary examination to determine the effect of DMV stimulation on the cognitive functions in elderly subjects with moderate dementia. In that single-arm study conducted on five participants aged over 85 years old with AD who had been treated with DMV stimulation at 15–40 Hz, we found that the DMV stimulation might have a positive impact on the memory function in older adults with moderate dementia. Moreover, in this study, we compared that 15–40 Hz DMV stimulation improved the cognitive functions in more larger sample size (35 elderly subjects) with MCI or mild dementia. Previous studies have examined the effect of exposure of AD patients to somatosensory stimulation at 40 Hz. The stimulation was in the form of rhythmic sensory stimulation (RSS), which works by deep stimulation of the mechanoreceptors. The results indicate that RSS may have a beneficial therapeutic effect in patients with AD [16,30]. DMV stimulation is a very low, inaudible sound below 40 Hz, which is a bass sound that belongs to the inaudible range to the human ear [26]. The 40-Hz oscillations of DMV stimulation suggest that they are present at many levels of the central nervous system and that vibrotactile stimulation can produce cognition in humans [31]. Furthermore, previous studies have been shown that 40 Hz stimuli appear to be involved in brain communication in general, may stimulate spontaneous generation which decreases with the onset of AD, and induce gamma responses in auditory and somatosensory stimulation, making it a suitable frequency for brain stimulation in AD patients and the basis for the use 15–40 Hz DMV stimulation in this study [32,33,34,35]. 

In the current study, we found improved performance in the WM test, and improved executive functions and information processing speed in the participants after 4 weeks of continuous DMV stimulation. Furthermore, our findings suggested differential effects of DMV stimulation between the MCI group and mild dementia group. It has been reported that acoustic stimulation at 40 Hz improved the cognitive functions in AD-transgenic mice (5XFAD). The results indicated that sounds at 40 Hz induce gamma oscillations in both the auditory cortex and the hippocampus [36,37]. Recently, stimulation at 40 Hz has been examined in humans and been reported to induce a wide range of neuronal entrainment [38]. The WM used in this study consisted of recognition and delayed memory, therefore our results lend support to the hypothesis that auditory stimulation entrains the hippocampal neurons to improve memory.

Next, we newly discovered that 15–40 Hz DMV stimulation can also improve executive functions and information processing speed, while previous studies have shown improvements in the memory, quality of sleep, and mental functioning [16,39,40]. Only a few non-pharmacological treatments have been shown to improve the executive function and information processing speed. Executive function is the ability required to plan, organize, operate on working memory, and switch between tasks, which are known to require many distinct brain regions working in tandem in order for complex tasks to be efficiently accomplished [41]. A previous study reported that aerobic exercises increase the connectivity in the frontal lobe network and promote improved motor performance in elderly people [42]. It was also reported that the frontoparietal network may be an essential area of intersection between the motor and executive functions, allowing for multimodal physical training to improve the executive functions in elderly people with MCI [43]. Thus, higher executive functions by DMV might not only reflect improvement of the targeted cognitive functions, but also accurate fast responses. As elderly people usually suffer from worsened cognitive and motor functions, our results indicated that early-stage intervention might be important for achieving a reliable benefit; there was a significant improvement of the executive functions in the MCI group, but not in the mild dementia group after DMV. On the other hand, it has been argued that the information processing speed is a fundamental characteristic of the brain’s cognitive efficiency. Performance in simple tests of processing speed is associated with the scores in more complex cognitive tests, such as reasoning [44]. Previous research has shown that the information processing speed is strongly related to the executive functions [45]. The link with information processing speed may be natural, given the need to switch between numbers and letters as quickly as possible when performing TMT-B.

Finally, a difference in the effectiveness of DMV stimulation on the performance of the subjects in TMT-A and TMT-B was observed. TMT is one of the most commonly used tests for assessing the executive functions in clinical neuropsychological assessment. TMT-A is often conducted as a baseline measure of motor and visual search speed, while TMT-B is used as a measure of set-shift and inhibition. The results of this study showed a significant improvement of the performance in TMT-B, but not in TMT-A, of the participants overall and in the MCI group. It is plausible that the worse executive functions in the MCI group at the baseline allowed greater room for improvement.

The limitations of the present study need to be taken into account for conducting further research. In this study, sample sizes were small, moreover, the majority of the subjects were women. In addition, differences in the degree of improvement in performance between the MCI and mild dementia groups could not be pursued. In the future, we should perform an additional study to clarify the potential differences (e.g., subtypes of dementia) and new research methods (e.g., crossover trials), such as add a control group.

## 5. Conclusions

Our findings of this study suggest that 15–40 Hz DMV stimulation improved the memory, executive functions, and information processing speed in elderly people with MCI and mild dementia. Furthermore, DMV stimulation at 15–40 Hz is might be effective for improving impaired cognition, particularly in elderly people with MCI.

## Figures and Tables

**Figure 1 ijerph-19-03803-f001:**
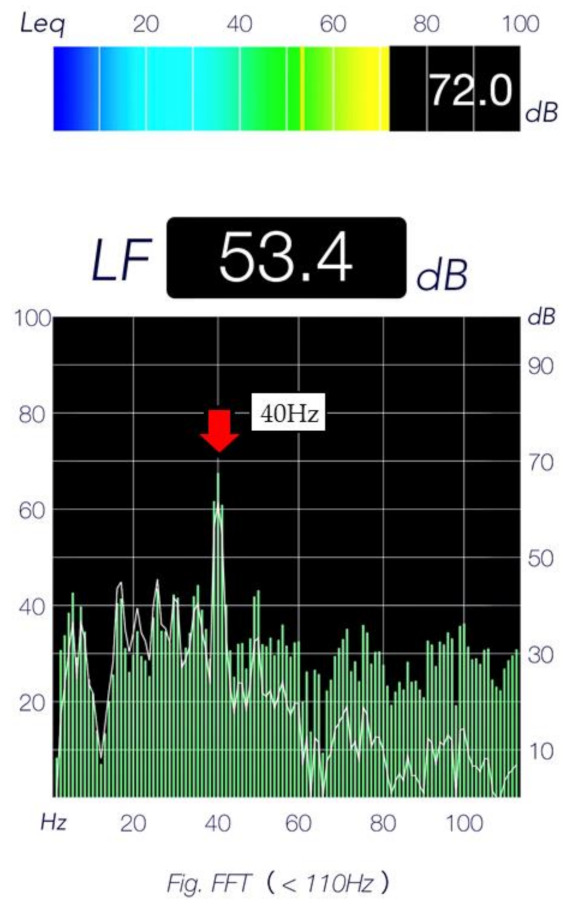
Acoustic analysis software to perform acoustic analysis.

**Figure 2 ijerph-19-03803-f002:**
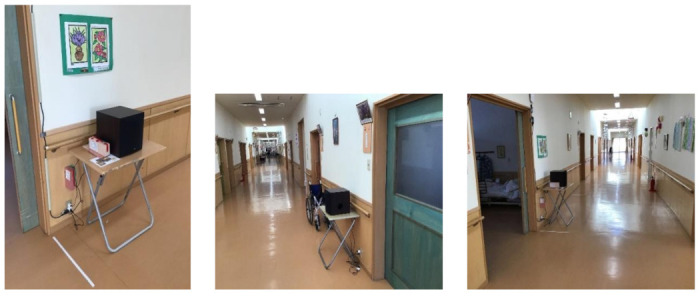
Installation view of the device.

**Figure 3 ijerph-19-03803-f003:**
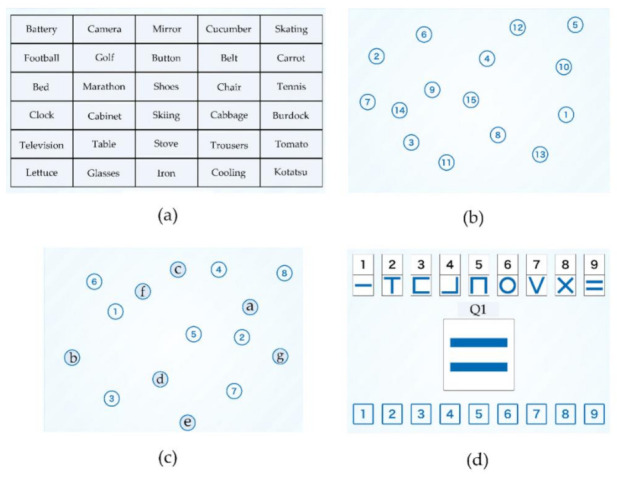
Example of NCGG-FAT. (**a**) Subject 1: Tablet Version of Word list Memory. (**b**) Subject 2, Tablet Version of Trail Making Test Version A. (**c**) Subject 3, Tablet Version of Trail Making Test Version B. (**d**) Subject 4: Symbol Digit Substitution Task.

**Table 1 ijerph-19-03803-t001:** Characteristics of participants.

	MCI Group		Mild Dementia Group			
	*n* = 17		*n* = 18			
	Mean	SD	Mean	SD	*p* Value	95% CI
Age (years)	84.6	9	88.3	6.4	0.171	80.23, 92.74
Gender (% female)	76		89		0.076	
Height (cm)	140.4	6.9	140.8	7.9	0.893	136.71, 144.46
Weight (kg)	46.4	7.8	47.4	8.7	0.747	42.61, 51.20
BMI (kg/m^2^)	22.7	2.9	24.7	3.9	0.094	21.75, 25.76
BI (score)	68.5	16.7	58.1	26.4	0.227	54.76, 73.38
NPI-NH (score)	9.8	12.9	22.9	20	0.037 *	10.85, 20.09

* *p* < 0.05, unpaired *t*-test. BI: Barthel Index; NPI-NH: Neuropsychiatric Inventory-Nursing Home edition.

**Table 2 ijerph-19-03803-t002:** Result of pre-test and post-test in the participants.

	Pre-Test		Post-Test		*p* Value	95% CI
	Mean	Mean	Mean	Mean		
GS (kg)	12.5	6.2	13.4	5.4	0.555	8.61, 18.48
UWS (m/s)	0.69	0.18	0.61	0.29	0.582	0.44, 0.71
WM (score)	3.8	2.6	4.7	2.5	0.021 *	2.44, 5.97
TMT-A (s)	104.1	88.4	89.1	82.5	0.761	70.78, 123.53
TMT-B (s)	209.8	96.0	166.2	92.3	0.036 *	125.32, 254.03
SDST (score)	9.4	7.6	11.6	7.0	0.006 **	4.46, 16.30
MMSE (score)	18.4	6.4	19.2	5.8	0.425	15.81, 21.79

* *p* < 0.05, ** *p* < 0.01, paired *t*-test. GS: Grip strength; UWS: Usual walking speed: WM: Word list memory; TMT-A: Trail making test-part A; TMT-B: Trail making test-part B; SDST: Symbol digit substitution task; MMSE: Mini-Mental State Examination.

**Table 3 ijerph-19-03803-t003:** Result of pre-test and post-test in each group.

	MCI Group					Mild Dementia Group				
	Pre-Test		Post-Test			PRE-TEST		Post-Test		
	Mean	SD	Mean	SD	*p* Value	Mean	SD	Mean	SD	*p* Value
GS (kg)	13.7	7.2	14.8	6.4	0.658	10.9	4.5	11.5	2.8	0.670
UWS (m/s)	0.69	0.2	0.56	0.34	0.876	0.6	0.24	0.66	0.26	0.436
WM (score)	4.8	2.9	5.7	3	0.259	2.8	1.9	3.6	1.4	0.015 *
TMT-A (s)	77.5	69.3	84.3	90.1	0.733	129.2	98.6	93.8	77.0	0.377
TMT-B (s)	219.8	102.9	167	105	0.037 *	200.4	90.9	165.3	81.5	0.491
SDST (score)	11	9	13.9	8.6	0.003 **	7.8	5.8	9.2	4.0	0.354
MMSE (score)	21.5	5.2	22.3	3.8	0.878	14.4	5.7	14.7	5.4	0.229

* *p* < 0.05, ** *p* < 0.01, paired *t*-test. GS: Grip strength; UWS: Usual walking speed: WM: Word list memory; TMT-A: Trail making test-part A; TMT-B: Trail making test-part B; SDST: Symbol digit substitution task; MMSE: Mini-Mental State Examination.

## Data Availability

Data available on request due to restrictions, e.g., privacy or ethical. The data presented in this study are available on request from the corresponding author.

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
