# Peer review of "The Effect of Deep Micro Vibrotactile Stimulation on Cognitive Function of Mild Cognitive Impairment and Mild Dementia"

_ijerph, 2022, doi:10.3390/ijerph19073803_

Round 1

Reviewer 1 Report

The English need a revision, starting from the abstract,  first of all in grammar and clarity.

Additionally, I suggest the author focus the improvement they report on processes related to attention and short-term verbal learning in laboratory tasks.

Page 2

line 1 please specify the type of performance mentioned in this line

line 57  check grammar and repetitions

line 58 : better to specify the goal: on a larger group? With given conditions?

Paragraph 2.1 includes both participants and procedure for the assessment: better to divide into two distinct paragraphs. Please,  add  here also a definition of MCI, given that  participants are divided into two groups

No mention is made of the diagnostic criteria,   nor a definition is provided of the processes that according to the authors are targeted by the intervention proposed.

Page 3

It is not always clear when the authors are mentioning outcomes with rats and when they refer to their study with humans.  I suggest focusing on previous studies with humans

Line 139 Rather than outcome, the heading of paragraph 2.3 should be 'Outcome measures'. This is what the readers see in the paragraph

Authors should check anytime they introduce a new acronym, they add the full name.

Page 4

Line 156: better refer to the process or the measure ( not Subject: it is confounding!) and use the full name of the process, given the relevance they have in the study.

Page 5

The description of the results is not always clear and easy to follow. This section in particular needs revision for the English language.

Line 193. Authors mix together information from statistical analysis and interpretation. The functions they refer to  should be  clearly addressed earlier where they describe the goals of the study  and however the comment made here should be moved ( and better described) in the discussion section

Line 198 on

Table 1 here is confounding: it should be moved earlier.

Page 6

Discussion

Ore effectively authors should start summarizing the goal ( not the results),  results from previous studies then describe what makes the current study relevant.

The authors should more clearly restrict the focus of their analysis to the specific components addressed and not generally mention “WM, executive function” besides information processing speed.

Additionally, readers of the journal may benefit from a clear description of these processes in the introduction ( adding relevant and recent references).  Authors should describe what are the specific processes tapped by the tasks,   with particular attention to tasks where they find a difference ( for instance by the  Trail Making Test). They should also comment why in their view the other tasks did not show differences and comment on the differences in the increased performance they found in the two groups, which might be relevant for  motivating any use of the intervention proposed and to highlight goals for further studies.  The conclusion should be then reformulated.

Author Response

Response to Reviewer 1 Comments

Point 1: The English need a revision, starting from the abstract,  first of all in grammar and clarity.

Response 1: Thank you for your supportive comments. Our manuscripts had been checked by a  native English speaker.

Point 2Additionally, I suggest the author focus the improvement they report on processes related to attention and short-term verbal learning in laboratory tasks. 

Response 2: Thank you for supportive comments. As you suggested, we have added more information of the improvement of TMT and SDST on Discussion as follows;

(Thus, higher executive functions by DMV might not only reflect improvement of the targeted cognitive functions, but also accurate fast responses. Because elderly people usually suffer from worsened cognitive and motor functions, our results indicated that early stage intervention might be an important for achieving a reliable benefit; there was a significant improvement of the executive functions in the MCI group, but not in the mild dementia group after DMV. Page8, Line290)

Point 3: line 1 please specify the type of performance mentioned in this line

Response 3: Thank you for supportive reviews comments. According to reviewer’s suggestion, we have revised and changed as following.

 (The 2 sources of stimulation were combined to provide stimulation at specific frequencies in the range of 1 to 30 Hz. The results showed an improvement in delayed memory in the subject group as compared with the control group. Page2, Line56)

Point 4: line 57 check grammar and repetitions

Response 4: Thank you for your supportive comments. According to reviewer’s suggestion, we have collected the grammar in line 57.

Point 5: line 58: better to specify the goal: on a larger group? With given conditions?

Response 5: Thank you for supportive comments. As the reviewer suggested, we have added "with given conditions" in line 58 as follow.

(Thus, the purpose of this study was to clarify the effects of 15-40 Hz DMV stimulation on the cognitive functions in elderly people with MCI or mild dementia with given conditions. Page2, Line72)

Point 6: Paragraph 2.1 includes both participants and procedure for the assessment: better to divide into two distinct paragraphs. Please, add here also a definition of MCI, given that participants are divided into two groups

Response 6: Thank you for supportive comments. As the reviewer suggested, we have divided into two distinct paragraphs, and we have described a definition of MCI in Paragraph 2.1.

(People with dementia experience decline in social and occupational functioning, and with disease progression, these patients become unable to take care of themselves more than 6 months, requiring (a) ongoing support for even the simplest activities of daily living, and (b) specific forms of support to replace abnormal behaviours (such as wandering) with more socially acceptable ones [1, 2]. The initial symptoms tend to worsen over time. They may be restricted to new and unfamiliar environments at first, eventually spreading to even familiar environments, with impaired spatial and geographical orientation and negative effects on the autonomy, independence, and self-confidence [3,4]. Page1, Line34)

Point 7: No mention is made of the diagnostic criteria, nor a definition is provided of the processes that according to the authors are targeted by the intervention proposed.

Response 7: Thank you for supportive reviews comments. According to reviewer’s suggestion, we have revised and changed as following;

(Furthermore, the CDR has been neuropathologically confirmed, and the content and criterion validity of this scale have been established [23, 24]. In the presence of dementia, the CDR can be used to monitor the entire course of the disease, from very mild to severe, avoiding floor and ceiling effects. Based on information obtained from interviews and cognitive functioning tests, each of the six cognitive and functional domains (memory, disorientation, judgement and problem solving, community activities, home and hobbies, and personal care) is rated on a 5-point scale, with 0 representing no impairment, 0.5, 1, 2 and 3 representing very mild, mild, moderate and severe dementia, respectively. The participants were divided into a mild cognitive impairment (MCI) group (CDR score 0.5) and a mild dementia group (CDR score 1). MCI was defined as follows for this study: 1) the person is neither normal nor has dementia; 2) there is evidence of cognitive deterioration shown by either objectively measured decline over time or subjective reporting of decline by the patient him/herself or informant, in conjunction with objective cognitive deficits; and 3) activities of daily living are preserved and complex instrumental functions are either intact or minimally impaired [25]. Page3, Line107)

Point8: It is not always clear when the authors are mentioning outcomes with rats and when they refer to their study with humans.  I suggest focusing on previous studies with humans

Response8Thank you for supportive comments. As the reviewer suggested, we had referenced previous study are as follows;

(When mice with AD were exposed to 40 Hz, the accumulation of amyloid-beta was reduced and cognitive functions were improved, suggesting that 40 Hz may inhibit the progression of AD [26]. Hannah, F.I.; Annnabelle, C.S.; Anthony, J.M.; Andril, R.; Fan, G.; Tyler, Z.G.; et al. Gamma frequency entrainment attenuates amyloid and modifies microglia. Nature 2016, 540(7632): 230-235.)

Point9Line 139 Rather than outcome, the heading of paragraph 2.3 should be 'Outcome measures'. This is what the readers see in the paragraph

Response9Thank you for supportive reviews comments. According to reviewer’s suggestion, we have revised and changed as “Outcome measures” in paragraph 2.4.

Point10Authors should check anytime they introduce a new acronym, they add the full name.

Response10Thank you for supportive reviews comments. According to reviewer’s suggestion, we have added the full name.

(We assessed the patients for changes in the grip strength (GS) and usual walking speed (UWS) after the intervention. Page4, Line174)

Point11: Line 156: better refer to the process or the measure (not Subject: it is confounding!) and use the full name of the process, given the relevance they have in the study. 

Response11Thank you for supportive reviews comments. According to reviewer’s suggestion, we have revised and changed in Line 156.

(“The WM test consists” Page5, Line190)

Point12The description of the results is not always clear and easy to follow. This section in particular needs revision for the English language.

Response12Thank you for supportive comments. We have collected the English language of the results as follows;

(The unpaired t-test was used to analyse the differences in the characteristics of the participants between the two groups, and revealed a significant difference in the NPI-NH score between the two groups (p < 0.05). However, there were no differences in any of the subject characteristics, such as the age, gender distribution, height, weight, BMI, or BI at the baseline between the two groups.

 Next, a comparison of the pre-test and post-test results in the participants is shown in Table 2 and Table 3. The paired t-test was used to analyse the pre-test/post-test differences, and revealed significant differences in the results of the WM test (p < 0.05), TMT-B (p < 0.05), and SDST (p < 0.01). Page6, Line224)

Point13Line 193. Authors mix together information from statistical analysis and interpretation. The functions they refer to should be clearly addressed earlier where they describe the goals of the study and however the comment made here should be moved (and better described) in the discussion section

Response13Thank you for supportive comments. As the reviewer suggested, we have removed the interpretation of statistical analyses from this section.

Point14Line 198 on Table 1 here is confounding: it should be moved earlier.

Response14Thank you for supportive comments. As you suggested, we have moved earlier Table 1.

Point15Ore effectively authors should start summarizing the goal ( not the results),  results from previous studies then describe what makes the current study relevant.

Response15Thank you for supportive comments. As the reviewer suggested, we have restructured the foreword to the discussion.

(In our previous study, we conducted a preliminary examination to determine the effect of DMV stimulation on the cognitive functions in elderly subjects with moderate dementia. In that single-arm study conducted on 5 participants aged over 85 years old with AD who had been treated with DMV stimulation at 15-40 Hz, we found that the DMV stimulation might have a positive impact on the memory function in older adults with moderate dementia. Moreover, in this study, we compared that 15-40 Hz DMV stimulation improved the cognitive functions in more larger sample size (35 elderly subjects) with MCI or mild dementia. Previous studies have examined the effect of exposure of AD patients to somatosensory stimulation at 40 Hz. Page7, Line247)

(In the current study, we found improved performance in the WM test, and improved executive functions and information processing speed in the participants after 4 weeks of continuous DMV stimulation. Furthermore, our findings suggested differential effects of DMV stimulation between the MCI group and mild dementia group. Page7, Line267)

Point16The authors should more clearly restrict the focus of their analysis to the specific components addressed and not generally mention “WM, executive function” besides information processing speed.

Response16Thank you for supportive comments. As the reviewer suggested, we have added discussion focused to on the characteristics of WM and TMT-B.

(It has been reported that acoustic stimulation at 40 Hz improved the cognitive functions in AD-transgenic mice (5XFAD). The results indicated that sounds at 40 Hz induce gamma oscillations in both the auditory cortex and the hippocampus [37, 38]. Recently, stimulation at 40 Hz has been examined in humans and been reported to induce a wide range of neuronal entrainment [39]. The WM used in this study consisted of  recognition and delayed memory, therefore our results lend support to the hypothesis that auditory stimulation entrain the hippocampal neurons to improve memory. Page7, Line270)

(Executive function is the ability required to plan, organize, operate on working memory, and switch between tasks, which are known to require many distinct brain regions working in tandem in order for complex tasks to be efficiently accomplished [42]. Page8, Line282)

(Thus, higher executive functions by DMV might not only reflect improvement of the tar-geted cognitive functions, but also accurate fast responses. Because elderly people usually suffer from worsened cognitive and motor functions, our results indicated that early stage intervention might be an important for achieving a reliable benefit; there was a significant improvement of the executive functions in the MCI group, but not in the mild dementia group after DMV. Page8. Line290)

Point17Additionally, readers of the journal may benefit from a clear description of these processes in the introduction ( adding relevant and recent references).  Authors should describe what are the specific processes tapped by the tasks, with particular attention to tasks where they find a difference ( for instance by the  Trail Making Test). They should also comment why in their view the other tasks did not show differences and comment on the differences in the increased performance they found in the two groups, which might be relevant for motivating any use of the intervention proposed and to highlight goals for further studies.  The conclusion should be then reformulated.

Response17Thank you for supportive comments. As the reviewer suggested, we have added information for our previous study [17], and have revised and changed the discussion. We are so sorry, but we could not clarify the mechanism about the differences performance between the two groups. Therefore, we should perform the examination with adequate control group in the near future.

(In our previous study, we reported that DMV stimulation at 15-40 Hz may be useful as one of the non-pharmacological treatments to enhance short-term memory in AD patients, however, was quite small-sample size (5 participants) [17]. Page2, Line68)

(Finally, a difference in the effectiveness of DMV stimulation on the performance of the subjects in TMT-A and TMT-B was observed. TMT is one of the most commonly used tests for assessing the executive functions in clinical neuropsychological assessment. TMT-A is often conducted as a baseline measure of motor and visual search speed, while TMT-B is used as a measure of set-shift and inhibition. The results of this study showed a significant improvement of the performance in TMT-B, but not in TMT-A, of the participants overall and in the MCI group. It is plausible that the worse executive functions in the MCI group at the baseline allowed greater room for improvement. Page8, Line302)

(In the future, we should perform an additional study to clarify the potential differences (e.g. subtypes of dementia) and new research methods (e.g. crossover trials), add a control group. Page8, Line313)

Reviewer 2 Report

The ms. entitled "The Effect of Deep Micro Vibrotactile stimulation on cognitive function of MCI and mild dementia" is interesting, but some issues are present

Please use mild cognitive impairment in the title instead of MCI

Review the abstract: Please add the take home message

Review the introduction: The introduction is lacking of information about mild cognitive impairment and its manifestations, and the transition from mild cognitive impairement to AD (e.g.,10.3109/17518423.2012.749951).

Moreover the aim and the hypotheses need to be clarified. Finally at the end of the introduction the authors say "elderly people with mild and 
moderate dementia": is MCI the same of mild dementia? 

Review the results: How did the authors determine the sample size?

Is the MCI group the same of moderate dementia? The authors should be consistent throught the manuscript, using the same labels.

Why is the control group not present?

Please check some typos throught the manuscript

Please specify the meaning of mild AD?

Author Response

Response to Reviewer 2 Comments

Point 1Review the abstract: Please add the take home message

Response 1: Thank you for your comments. As the reviewer suggested, we have added take home message.

(Furthermore, our novel findings showed the different effectiveness of the treatment depending on the stage of cognitive impairments. Page1, Line27)

Point 2: Review the introduction: The introduction is lacking of information about mild cognitive impairment and its manifestations, and the transition from mild cognitive impairment to AD (e.g.,10.3109/17518423.2012.749951).

Response 2: Thank you for supportive reviews comments. According to reviewer’s suggestion, we have revised preface of introduction.

(With the ageing of the population in Japan, the prevalence of dementia is increasing at an alarming rate. People with dementia experience decline in social and occupational functioning, and with disease progression, these patients become unable to take care of themselves more than 6 months, requiring (a) ongoing support for even the simplest activ-ities of daily living, and (b) specific forms of support to replace abnormal behaviours (such as wandering) with more socially acceptable ones [1, 2]. The initial symptoms tend to worsen over time. They may be restricted to new and unfamiliar environments at first, eventually spreading to even familiar environments, with impaired spatial and geograph-ical orientation and negative effects on the autonomy, independence, and self-confidence [3,4]. Page1, Line33)

Point 3: Moreover the aim and the hypotheses need to be clarified. Finally at the end of the introduction the authors say "elderly people with mild and moderate dementia": is MCI the same of mild dementia? 

Response 3: Thank you for supportive reviews comments. According to reviewer’s suggestion, we have revised and changed as following.

(Thus, the purpose of this study was to clarify the effects of 15-40 Hz DMV stimulation on the cognitive functions in elderly people with MCI or mild dementia with given conditions. Page2, Line72)

(Furthermore, previous studies have been shown that 40 Hz stimuli appear to be involved in brain communication in general, may stimulate spontaneous generation which decreases with the onset of AD, and induce gamma responses in auditory and somatosensory stimulation, making it a suitable frequency for brain stimulation in AD patients and the basis for the use 15-40 Hz DMV stimulation in this study [33-36]. Page7, Line262)

Point 4: Review the results: How did the authors determine the sample size?

Response 4: Thank you for your comments. As you suggested, we have added about information related to the sample size of the subjects. (2. Materials and Methods 2.1 Participants Page2, Line75)

Point 5: Is the MCI group the same of moderate dementia? The authors should be consistent throught the manuscript, using the same labels.

Response 5: Thank you for your supportive comments. As you suggested, we made mistakes about the term of “moderate” dementia. So we have collected “moderate” to “mild” and, we have described the difference between the term of “MCI” and “mild” dementia.

Point 6: Why is the control group not present?

Response 6: Thank you for your supportive comments. We have conducted a single arm intervention trial in this study, but as you suggested, we should perform a case-control study design in the near future.

(In the future, we should perform an additional study to clarify the potential differences (e.g. subtypes of dementia) and new research methods (e.g. crossover trials), add a control group. Page 8, Line 313)

Point7: Please check some typos throught the manuscript

Response7Thank you for your supportive comments. According to your suggestions, our manuscripts had been checked all typos by a native English speaker.

Point8Please specify the meaning of mild AD?

Response8Thank you for your supportive comments. We have described the definition of mild AD as follows;

(Furthermore, the CDR has been neuropathologically confirmed, and the content and criterion validity of this scale have been established [23, 24]. In the presence of dementia, the CDR can be used to monitor the entire course of the disease, from very mild to severe, avoiding floor and ceiling effects. Based on information obtained from interviews and cognitive functioning tests, each of the six cognitive and functional domains (memory, disorientation, judgement and problem solving, community activities, home and hobbies, and personal care) is rated on a 5-point scale, with 0 representing no impairment, 0.5, 1, 2 and 3 representing very mild, mild, moderate and severe dementia, respectively. The participants were divided into a mild cognitive impairment (MCI) group (CDR score 0.5) and a mild dementia group (CDR score 1). MCI was defined as follows for this study: 1) the person is neither normal nor has dementia; 2) there is evidence of cognitive deterioration shown by either objectively measured decline over time or subjective reporting of decline by the patient him/herself or informant, in conjunction with objective cognitive deficits; and 3) activities of daily living are preserved and complex instrumental functions are either intact or minimally impaired [25]. Page3, Line107)

Reviewer 3 Report

The paper is well presented in all aspects of form and includes the correct sections of a scientific paper. The data and results are presented in an appropriate way. However, there are some aspects to improve:

  • The title includes an abbreviation that is not clear and is not described throughout the text. May lead to confusion between mild or moderate degrees of dementia or cognitive impairment.
  • Some assessment tools are cited and described extensively, and in different sections of the paper. It is recommended to abbreviate, clarify and unify all the descriptions of the used tools in a section, different from "participants".
  • Some of the assessment tools are not included in the results or discussion.
  • On line 139, the title should be outcomes measures.
  • The study design is not described.
  • The description of the functional and cognitive characteristics of the sample can improve.
  • The description of statistical analysis should be improved.
  • Table 1 is not mentioned in the text.
  • Significant results in table 1 are not highlighted.
  • There is an errors in the SDST units of measurement in tables 2 and 3.
  • The discussion needs to be improved, with a deeper analysis and better support for the results obtained, especially the statements on memory, executive function and processing speed. Discuss the limitations of the technique.
  • Some limitations of the study are described, but not other important ones, such as the sample size.

Appropriate references are cited in the text and the reference list is suitable, recent and balanced.

Author Response

Response to Reviewer 3 Comments

Point 1: The title includes an abbreviation that is not clear and is not described throughout the text. May lead to confusion between mild or moderate degrees of dementia or cognitive impairment.

Response 1: Thank you for your supportive reviews comments. According to your suggestion, we have revised and changed as follows. In addition, as you suggested we made mistakes about the term of “moderate” dementia. So, we have collected "moderate" to "mild", and we have described the difference between the term of “MCI” and “mild dementia”.

(The Effect of Deep Micro Vibrotactile stimulation on cognitive function of mild cognitive impairment and mild dementia: title)

Point 2Some assessment tools are cited and described extensively, and in different sections of the paper. It is recommended to abbreviate, clarify and unify all the descriptions of the used tools in a section, different from "participants".

Response 2: Thank you for your supportive comments. According to your suggestion, we have revised “procedure for the assessment”.

(2. Materials and Methods 2.1 Participants Page2, Line75)

Point 3: Some of the assessment tools are not included in the results or discussion.

Response 3: Thank you for your supportive comments. According to your suggestion, we have revised and described about the assessment tools Results and Discussion.

(Thus, higher executive functions by DMV might not only reflect improvement of the tar-geted cognitive functions, but also accurate fast responses. Because elderly people usually suffer from worsened cognitive and motor functions, our results indicated that early stage intervention might be an important for achieving a reliable benefit; there was a significant improvement of the executive functions in the MCI group, but not in the mild dementia group after DMV. On the other hand, it has been argued that the information processing speed is a fundamental characteristic of the brain’s cognitive efficiency. Performance in simple tests of processing speed is associated with the scores in more complex cognitive tests, such as reasoning [45]. Previous research has shown that the information processing speed is strongly related to the executive functions [46]. The link with information processing speed may be natural, given the need to switch between numbers and letters as quickly as possible when performing TMT-B. Finally, a difference in the effectiveness of DMV stimulation on the performance of the subjects in TMT-A and TMT-B was observed. TMT is one of the most commonly used tests for assessing the executive functions in clinical neuropsychological assessment. TMT-A is often conducted as a baseline measure of motor and visual search speed, while TMT-B is used as a measure of set-shift and inhibition. The results of this study showed a significant improvement of the performance in TMT-B, but not in TMT-A, of the participants overall and in the MCI group. It is plausible that the worse executive functions in the MCI group at the baseline allowed greater room for improvement. Page8, Line290)

Point 4: On line 139, the title should be outcomes measures.

Response 4: Thank you for your supportive comments. According to your reviewer’s suggestion, we have changed to " outcomes measures".

Point 5: The study design is not described.

Response 5: Thank you for supportive comments. As the reviewer suggested, we have described " the study design”.

(Therefore, we conducted a single intervention trial with DMV stimulation for 35 participants Page2, Line 85)

Point 6: The description of the functional and cognitive characteristics of the sample can improve.

Response 6: Thank you for supportive reviews comments. According to reviewer’s suggestion, we have revised and changed the description of the functional and cognitive characteristics.

(The target population was elderly people aged 65 years or over who were using Japanese welfare services for elderly people. Subjects with mood/ anxiety disorders, mental retardation, and/or severe systemic diseases were excluded. Since the main purpose of this study was to demonstrate a response to DMV stimulation to inform the next proof of concept study, representation of different stages of cognitive dysfunction was crucial. Page2, Line78)

Point 7: The description of statistical analysis should be improved.

Response 7: Thank you for your supportive comments. According to your suggestion, we have revised and changed the description of the functional and cognitive characteristics of sample.

(The unpaired t-test was used to analyze the differences in the characteristics of the participants between the two groups, and revealed a significant difference in the NPI-NH score between the two groups (p < 0.05). However, there were no differences in any of the subject characteristics, such as the age, gender distribution, height, weight, BMI, or BI at the baseline between the two groups.  Next, a comparison of the pre-test and post-test results in the participants is shown in Table 2 and Table 3. The paired t-test was used to analyze the pre-test/post-test differences, and revealed significant differences in the results of the WM test (p < 0.05), TMT-B (p < 0.05), and SDST (p < 0.01). Moreover, we also compared the pre-test/post-test differences in the results between the two groups (Table 3). Page6, Line224)

Point8: Table 1 is not mentioned in the text.

Response8Thank you for your supportive comments. As you suggested, we have moved “Table 1”.

(Page6, Line219)

Point9Significant results in table 1 are not highlighted.

Response9Thank you for your comments. As you suggested, we have highlighted the significant results in Table 1. (Page6, Line219)

Point10There is an errors in the SDST units of measurement in tables 2 and 3.

Response10Thank you for your supportive comments. According toyour suggestion, we have revised and changed the SDST units of measurement in Tables 2 and 3.

Point11The discussion needs to be improved, with a deeper analysis and better support for the results obtained, especially the statements on memory, executive function and processing speed. Discuss the limitations of the technique.

Response11Thank you for your supportive comments. According to your suggestion, we have revised the Discussion.

(Thus, higher executive functions by DMV might not only reflect improvement of the tar-geted cognitive functions, but also accurate fast responses. Because elderly people usually suffer from worsened cognitive and motor functions, our results indicated that early stage intervention might be an important for achieving a reliable benefit; there was a significant improvement of the executive functions in the MCI group, but not in the mild dementia group after DMV. On the other hand, it has been argued that the information processing speed is a fundamental characteristic of the brain’s cognitive efficiency. Performance in simple tests of processing speed is associated with the scores in more complex cognitive tests, such as reasoning [45]. Previous research has shown that the information processing speed is strongly related to the executive functions [46]. The link with information processing speed may be natural, given the need to switch between numbers and letters as quickly as possible when performing TMT-B.

Finally, a difference in the effectiveness of DMV stimulation on the performance of the subjects in TMT-A and TMT-B was observed. TMT is one of the most commonly used tests for assessing the executive functions in clinical neuropsychological assessment. TMT-A is often conducted as a baseline measure of motor and visual search speed, while TMT-B is used as a measure of set-shift and inhibition. The results of this study showed a significant improvement of the performance in TMT-B, but not in TMT-A, of the participants overall and in the MCI group. It is plausible that the worse executive functions in the MCI group at the baseline allowed greater room for improvement. Page8, Line290)

Point12Some limitations of the study are described, but not other important ones, such as the sample size.

Response12: Thank you for your supportive comments. As you suggested, we have revised the Limitaiton.

(In this study, sample sizes were small, moreover, the majority of the subjects were women. In the future, we should perform an additional study to clarify the potential differences (e.g. subtypes of dementia) and new research methods (e.g. crossover trials), add a control group. Page 8 Line 311)

Round 2

Reviewer 2 Report

The authors replied to all the questions rased. The paper could be accepted as it is